# Characterization of a New Stripe Rust Resistance Gene on Chromosome 2StS from *Thinopyrum intermedium* in Wheat

**DOI:** 10.3390/plants14101538

**Published:** 2025-05-20

**Authors:** Chengzhi Jiang, Yujie Luo, Doudou Huang, Meiling Chen, Ennian Yang, Guangrong Li, Zujun Yang

**Affiliations:** 1School of Life Science and Technology, University of Electronic Science and Technology of China, Chengdu 610054, China; 202311140626@std.uestc.edu.cn (C.J.); 202321140401@std.uestc.edu.cn (Y.L.); 202221140532@std.uestc.edu.cn (D.H.); 202221140504@std.uestc.edu.cn (M.C.); 2Crop Research Institute, Sichuan Academy of Agricultural Sciences, Chengdu 610066, China; yangennian@126.com

**Keywords:** *Thinopyrum intermedium*, ND-FISH, molecular markers, stripe rust resistance

## Abstract

Stripe rust, caused by *Puccinia striiformis* f. sp. *tritici*, is a highly destructive disease prevalent across most wheat-growing regions globally. The most effective strategy for combating this disease is through the exploitation of durable and robust resistance genes from the relatives of wheat. *Thinopyrum intermedium* (Host) Barkworth and D.R. Dewey has been widely hybridized with common wheat and has been shown to be a valuable source of genes, conferring resistance and tolerance against both the biotic and abiotic stresses affecting wheat. In this study, a novel wheat–*Th. intermedium* 2StS.2J^S^L addition line, named Th93-1-6, which originated from wheat–*Th. intermedium* partial amphidiploid line, Th24-19-5, was comprehensively characterized using nondenaturing-fluorescence *in situ* hybridization (ND-FISH) and Oligo-FISH painting techniques. To detect plants with the transfer of resistance genes from Th93-1-6 to wheat chromosomes, 2384 M_1_-M_3_ plants from the cross between Th93-1-6 and the susceptible wheat cultivar MY11 were studied by ND-FISH using multiple probes. A total of 37 types of 2StS.2J^S^L chromosomal aberrations were identified. Subsequently, 12 homozygous lines were developed to construct a cytological bin map. Ten chromosomal bins on the 2StS.2J^S^L chromosome were constructed based on 84 specific molecular markers. Among them, eight alien chromosome aberration lines, which all contained the bin 2StS-3, showed enhanced stripe rust resistance. Consequently, the gene(s) for stripe rust resistance was physically mapped to the 92.88-155.32 Mb region of 2StS in *Thinopyrum intermedium* reference genome sequences v2.1. Moreover, these newly developed wheat–*Th. intermedium* 2StS.2J^S^L translocation lines are expected to serve as valuable genetic resources in the breeding of rust-resistant wheat cultivars.

## 1. Introduction

Common wheat (*Triticum aestivum* L., 2*n* = 6*x* = 42, AABBDD) is the most extensively cultivated cereal crop worldwide and contributes approximately 20% of the total consumed calories for humans [1,2]. Stripe rust, also known as yellow rust, caused by the fungus *Puccinia striiformis* f. sp. *tritici*, is a devastating airborne fungal disease of wheat. This disease exerts a significant negative impact on both the yield and quality of wheat [3,4]. Developing highly resistant cultivars is the most effective and environmentally friendly approach to address the challenges posed by this fungal disease. To date, 86 stripe rust resistance genes (*Yr1*–*Yr86*) have been formally cataloged. In addition, more than 100 provisionally designated genes and over 300 resistance loci associated with stripe rust resistance have been identified in common wheat and its relatives [5,6]. Nevertheless, most of these stripe rust resistance genes have been defeated because of the appearance and spread of new virulent rust pathotypes [7]. Consequently, there is an urgent need to continuously explore and utilize novel gene sources with broad-spectrum stripe rust resistance and incorporate these genes into wheat breeding programs.

*Thinopyrum intermedium* (Host) Barkworth and D.R. Dewey (JJJ^S^J^S^StSt, 2*n* = 6*x* = 42), a perennial Triticeae species related to hexaploid wheat, is known to be an important source of tolerance to salinity, drought, and resistance to the diseases of wheat [8,9,10]. Several disease resistance genes, such as those against leaf rust, stem rust, yellow rust, and powdery mildew (*Lr38*, *Sr44*, *Yr50*, *Pm40*, and *Pm43*) have been successfully introduced from *Th. intermedium* into wheat for resistance breeding purposes [11,12,13,14,15]. Recently, Luo et al. [16] bred a new wheat–*Th. intermedium* partial amphiploid line 92,048 carrying resistance to both stripe rust and *Fusarium* head blight (FHB). Yu et al. [17] developed a set of wheat–*Th. intermedium* addition lines by crossing wheat cultivar MY11 with wheat–*Th. intermedium* partial amphiploid line TAI7045 and line 78,784, and found that chromosomes 2St and 4St carried high resistance to stripe rust and powdery mildew. Subsequently, a series of wheat–*Th. intermedium*-derived lines carrying chromosomes 1St, 3St, 4J^S^, 5J^S^ and 7J^S^ showed high resistance to stripe rust [18,19,20,21,22]. Therefore, these potentially new resistance sources from *Th. intermedium* should be further characterized to develop effective and durable stripe rust resistance in wheat crops.

Genomic *in situ* hybridization (GISH) and fluorescence *in situ* hybridization (FISH) have been widely used to distinguish the J, J^S^ and St subgenomes and to detect *Th. intermedium* chromosomes in wheat backgrounds [23,24]. For example, Zhang et al. [25] characterized the *Th. intermedium* group-6 translocation T6StS.6J^S^L in line CH51 by GISH using total genomic DNA from *Pseudorogneria spicata* as a probe. Wang et al. [26] established standard karyotypes of *Th. intermedium* and its potential progenitor species using the FISH procedure. Nondenaturing FISH (ND-FISH) technology, combined with oligonucleotide (oligo) probes, was rapidly applied for the detection of alien chromosomes in wheat backgrounds [27]. Later, an Oligo-FISH painting system based on seven bulked pools of single-copy sequences was developed for the identification of Triticeae linkage groups [28]. This facilitated the documentation of complicated intra-genome translocations and the assignment of the alien chromosomes in different wheat–*Th. intermedium*-derived lines [29].

In the present study, we precisely characterized the genomic constitution of the wheat–*Th. intermedium* substitution line Th93-1-6 by Oligo-FISH painting and ND-FISH with multiple probes. In addition, extensive chromosomal rearrangements in an M_1_-M_3_ population of monosomic alien substitution lines were characterized by ND-FISH. We subsequently identified 12 homozygous wheat–*Th. intermedium* deletions and translocations that enabled physical mapping of the stripe rust resistance gene(s) on 2StS, which could accelerate transfer of the resistance into wheat backgrounds.

## 2. Results

### 2.1. Chromosomal Composition of Line Th93-1-6

ND-FISH analysis using probes Oligo-B11 + Oligo-pDb12H and Oligo-pSc119.2 + Oligo-pTa535 was conducted to characterize the *Th. intermedium* chromosomes in Th93-1-6. In our previous study, probe Oligo-pDb12H was able to detect the J^S^ chromosomes of *Th. intermedium*, while probe Oligo-B11 could distinguish the St and J chromosomes [29]. As shown in Figure 1a, Th93-1-6 contained 43 chromosomes, in which one pair of chromosomes carried Oligo-pDb12H signals on the long arms and strong Oligo-B11 hybridization on the short arms, which indicated that the long arms of J^S^ chromosomes had been connected to the short arms of St chromosomes to form the translocated chromosomes. Sequential ND-FISH with probes Oligo-pSc119.2 and Oligo-pTa535 revealed that Th93-1-6 had 41 wheat chromosomes with 2D absent, plus one more 1D chromosome. The alien chromosomes displayed relatively weak signals of probe Oligo-pTa535 at the terminal region (Figure 1b,c). Subsequently, the Oligo-FISH painting using probe Synt 2C generated distinct signals on the chromosome pairs of 2A, 2B and also on the two *Th. intermedium* chromosomes (Figure 1d). Based on the FISH patterns, we concluded that Th93-1-6 is a wheat–*Th. intermedium* 2StS.2J^S^L (2D) disomic substitution line. Meanwhile, we screened an additional four Oligo probes for the detection of chromosome 2StS.2J^S^L (Appendix A). For example, probe Oligo-pSc200 produced strong signals on the terminal regions of both the long and short arms (Appendix A). In addition, probe Oligo-6S111 displayed similar hybridization patterns to Oligo-pSc200; however, an extra Oligo-6S111 site was located on the sub-telomeric region of the long arm (Appendix A), which could be used to rapidly and effectively identify chromosome arms 2StS and 2J^S^L in the wheat background.

### 2.2. Chromosome Variations in M_1_ Population of Th93-1-6

A study was undertaken to investigate the transmission of the *Th. intermedium* 2StS.2J^S^L chromosomes in a wheat background. Th93-1-6 was used as the female parent in a cross with the wheat cultivar MY11; individual progeny plants were studied by ND-FISH. The standard karyotype of parent lines MY11 and Th93-1-6 by ND-FISH with probes Oligo-pSc119.2 and Oligo-pTa535 is shown in Appendix A. Chromosomes 4A and 5A exhibited FISH polymorphisms for the distribution of Oligo-pSc119.2 signals. A total of 1040 F_2_ seeds were obtained from the hybrids of MY11 and Th93-1-6; all the F_2_ seeds were irradiated with a dosage of 200 Gy (1.00 Gy/min). About 76.4% (795) of the seeds germinated and produced selfed progenies. Among the 795 plants, 268 (33.7%) plants contained monosomic 2StS.2J^S^L chromosomes, while 56 (7.0%) plants had two 2StS.2J^S^L chromosomes, indicating the intermediate transmission rate of 2StS.2J^S^L.

Subsequently, a total of 1823 M_1_ plants were recovered for karyotype analysis using probes Oligo-pSc119.2, Oligo-pTa535, Oligo-1RS-1 and Oligo-6S111. There were 431 (23.6%) individual plants that contained chromosome variations, in which 537 different types of translocations had occurred between the wheat chromosomes. The highest proportion of chromosome translocations involving the three wheat subgenomes was for B-D genomes, reaching 37.4%; the lowest was for A-A genome chromosomes at 3.2%. The typical FISH karyotypes of translocations are shown in Appendix A. Overall, 37 types of chromosomal aberrations involving 2StS.2J^S^L were identified, of which four carried deletions; 29 plants contained translocations (Figure 2). The remaining four variations were 2StS telosomes, 2J^S^L telosomes, iso-telosomic 2StS.2StS, and 2J^S^L.2J^S^L (Appendix A). These 2StS.2J^S^L deletions and translocations were crossed with the wheat cultivar MY11 for the later selection of homozygotes and the physical map construction.

### 2.3. Identification of Wheat–Th. intermedium 2StS.2J^S^L Deletions and Translocations

A total of 561 M_2_-M_3_ progenies derived from the 33 plants carrying aberrations involving the 2StS.2J^S^L chromosome were characterized by ND-FISH with multiple probes. Twelve plants (eight were disomic and four were monosomic) were characterized to have a single aberration type involving chromosome 2StS.2J^S^L. Among them, nine carried translocations and three contained deletions. Of the nine translocations, four were terminal translocations (1AS.1AL-2StS, W-7AS.7AL-2J^S^L, 2StS-6AS.6AL and 5DS.5DL-2J^S^L), and five were large-fragment translocations (1DS-2StS.2J^S^L, 2DS-2StS.2J^S^L, 2StS.2J^S^L-W, 7BS-2StS.2J^S^L and 2StS.2J^S^L-5DL). As shown in Figure 3a,b, line C671 had 41 wheat chromosomes and a pair of terminal translocated chromosomes. Sequential ND-FISH with probe Oligo-pSc119.2 + Oligo-pTa535 indicated that the wheat segment of this new translocation originated from chromosome 2D. This translocation chromosome in C671 was thus named 2DS-2StS.2J^S^L. Similarly, line C923 contained two partial 2StS.2J^S^L chromosomes translocated onto wheat chromosomes; probes Oligo-B11 + Oligo-K288, Oligo-pSc119.2 + Oligo-pTa535 revealed that 7BS was connected to the terminal region on chromosome 2StS.2J^S^L to form the translocation chromosome 7BS-2StS.2J^S^L (Figure 3c,d). Moreover, the deficiency of Oligo-1RS-1 signals indicated that the breakpoint was close to the centromeric regions. In addition, line C152 and C155 contained homozygous terminal translocations (Figure 3e–h). Sequential ND-FISH suggested that line C152 carried two 2StS-6AS.6AL chromosomes and that line C155 contained the complexly rearranged chromosome W-7AS.7AL-2J^S^L. The three deletions all involved the loss of fragments of the long arm. For example, line C367 contained 43 chromosomes with disomic 2StS.2J^S^L^del^-1 chromosome; sequential ND-FISH with probes Oligo-6S111 + Oligo-1RS-1 indicated that the breakpoint was between the two strong Oligo-6S111 sites (Figure 3i,j). Line C423 contained a single 2StS.2J^S^L^del^-2 chromosome, which had lost all Oligo-6S111 sites on the long arm (Figure 3k,l). The other deletions and translocations are shown in Appendix A.

### 2.4. Construction of a Physical Map of 2StS.2J^S^L Chromosomes

A total of 175 IT primers, 30 PLUG primers, 40 EST-PCR primers, and 6 newly developed primers were used to screen for 2StS.2J^S^L-specific PCR markers. Of these, 84 (70 IT primers, 7 PLUG primers, 4 EST-PCR primers, 3 newly developed primers) amplified specific bands associated with the *Th. intermedium* 2StS.2J^S^L chromosome, compared to wheat cultivar MY11. Thirty-five and 49 markers were distributed on chromosome arms 2StS and 2J^S^L, respectively. All the primer sequences were used for BLAST searching of the *Thinopyrum intermedium* reference genome sequences v2.1 by the B2DSC web sites to construct a physical map of 2StS.2J^S^L. Meanwhile, a total of 12 pairs of 2StS-specific primers showed polymorphic amplification between Th93-1-6 and Ps. spicata (St genome) and 16 pairs of primers produced different 2J^S^L-specific bands between Th93-1-6 and X24C-14 (2J^S^ addition lines), indicating that substantial sequence divergences had accumulated among the *Th. intermedium* group-2 chromosomes during the polyploidization.

Subsequently, all the 2StS.2J^S^L-specific markers were used to determine the breakpoints of the 12 homozygous 2StS.2J^S^L translocation and deletion lines. Among them, five translocation lines (C651, C671, C923, C42 and C152) included four different 2StS chromosomal bins. Line C651 contained bins 2StS-1-2StS-3 (92.88–232.65 Mb); Line C671 included bins 2StS-1-2StS-2 (151.15–232.65 Mb); Line C923 contained bin 2StS-1 (168.63–232.65 Mb); Line C42 and line C152 had the same breakpoints at bins 2StS-2-2StS-4 (0–174.64 Mb). The other five translocation lines and three deletion lines could divide 2J^S^L into six bins. The representative markers for different breakpoint types are shown in Figure 4. For example, 2StS.2J^S^L^del^-1 and 2StS.2J^S^L^del^-2 showed similar hybridization patterns with probes Oligo-pTa535 and Oligo-6S111; however, PCR amplification revealed that the distance of breakpoint positions between the two lines was more than 43.08 Mb. Thus, the markers could be anchored into ten cytological bins of chromosome 2StS.2J^S^L according to the amplification results: four for the short arm (2StS-1–2StS-4) and six for the long arm (2J^S^L-1–2J^S^L-6) (Figure 5). The constructed physical map was further used to map the stripe rust resistance locus on chromosome 2StS.2J^S^L.

### 2.5. Responses of Th93-1-6 and Derived Lines to Stripe Rust

In order to evaluate the contribution of the two *Th. intermedium* chromosome arms from Th93-1-6 to stripe rust resistance in the progeny of Th93-1-6 and MY11, we phenotyped the lines carrying 2StS telosomes, 2J^S^L telosomes, iso-telosomic 2StS.2StS and 2J^S^L.2J^S^L, as well as their parental lines, to *P. striiformis* f. sp. *tritici* races CYR32, CYR33, and CYR34 at both seedling and adult plant stages over the 2023–2024 and 2024–2025 growing season. As shown in Appendix A, Th93-1-6 was highly resistant to stripe rust (IT = 0) at both the seedling and adult stages, while MY11 was highly susceptible (IT = 4). Among their progenies, plants carrying *Th. intermedium* chromosome arm 2J^S^L were susceptible to stripe rust (IT = 3), whereas plants with arm 2StS were highly resistant to stripe rust (IT = 0). The results indicated that chromosome arm 2StS contributes to enhanced all-stage resistance (ASR) to stripe rust. We then randomly selected 30 individuals derived from the monosomic 2StS telosomic addition for stripe rust evaluation and molecular marker analysis using newly developed primers. The PCR amplification results indicated that 18 plants contained a 2StS-specific band and that all were highly resistant to stripe rust (Appendix A). The results provided additional evidence that this *Yr* locus contributes to stripe rust resistance.

We subsequently evaluated the adult-stage reactions of 12 of the 2StS.2J^S^L aberration lines. Four lines (C671, C923, C796, C115) were all stripe rust-susceptible, and eight lines (C651, C42, C152, C797, C423, C788, C269, C367) containing bin 2StS-3 were all stripe rust-resistant like the parental line, Th93-1-6. Thus, we further allocated the stripe rust-resistant locus to bin 2StS-3, which corresponded to the 92.88–155.32 Mb genome region in the *Th. intermedium* genome sequence of v2.1 (Figure 6).

### 2.6. Evaluation of Major Agronomic Traits of Th93-1-6 and Its Derived Lines

The yield performances of the two parental lines, Th93-1-6 and MY11, and also of the 2StS addition lines, 2J^S^L addition lines and the stripe rust-resistant translocation line C152, were investigated during the 2024–2025 growing season (Table 1, Appendix A). The plant height (PH), tiller number (TN) and 1000-grain weight (TGW) of Th93-1-6 were significantly higher than those of MY11. The TN and TGW of 2StS addition lines and line C152 were significantly higher than that of MY11; the 2J^S^L addition lines showed similar traits to MY11. In general, there were no significant negative impacts on the agronomic traits of the alien segments carrying stripe rust resistance, making it a valuable resource in wheat resistance breeding.

## 3. Discussion

Chromosomal rearrangements (CRs) play a crucial role in maintaining genome stability and enhancing environmental adaptability during speciation processes [30,31]. CRs have been extensively documented across diploid and polyploid Triticeae species. For example, the distal region of rye chromosome arm 6RL incorporates chromosomal segments of 3R and 7R [2]; *Elymus sibiricus* contains the major species-specific 4H/6H reciprocal translocation [32]. In addition, CRs also exist in *Thinopyrum* species. Chen et al. [33] employed Oligo-painting probes to identify 5E-7E and 4E-5E reciprocal translocations in various diploid *Th. elongatum* accessions. Subsequent GISH analyses by Chen et al. [23] revealed Robertsonian translocations between St and J^S^ chromosomes in *Th. intermedium*. Notably, while functioning as bridges for the transfer of valuable genes from *Th. intermedium* to wheat, wheat–*Th. intermedium* partial amphiploids also carry multiple CRs, including 1St-J^S^ (y70-1-4, TAI7047, TAI7045 and 78784); 5J-St (Zhong3-Zhong5, 8024, 78829 and 92048); 2St-J^S^ (Zhong2, Zhong4-Zhong5, 78784, 78829 and 8024) [16,29]. Significantly, these extensive CRs did not originate from the wide hybridization process between *Th. intermedium* and wheat, but rather pre-existed in various hexaploid *Th. intermedium* accessions prior to hybridization events. Moreover, chromosome translocations from *Th. intermedium* have been successfully introgressed into wheat for agronomic enhancement. Kruppa et al. [34] bred the *Thinopyrum* Robertsonian translocation 4StS.1J*^VS^* addition line GLA7, which conferred leaf and stripe rust resistance, drought tolerance, and yield stability. Similarly, line CH51, derived from TAI8335 and wheat cultivar Jintai 170, possessed resistance to the prevalent Chinese leaf rust and stripe rusts attributed to genes on chromosome 6StS.6J^S^L [25]. Furthermore, Larkin et al. [35] crossed Zhong5 and Wan 7107 and selected the 2Ai#2 (2St-J^S^) disomic addition line Z2, which was highly resistant to barley yellow dwarf virus and stripe rust. Yu et al. [17] generated a 2St-J^S^ addition line WT78-2 (78784/MY11 progeny) with dual resistance to stripe rust and powdery mildew. In our study, we characterized the karyotype of the wheat–*Th. intermedium* substitution line Th93-1-6 and found that chromosome 2StS.2J^S^L carried the stripe rust resistance. Though the three 2St-J^S^ translocations derived from different *Th. intermedium* accessions, the allelism of these stripe rust resistant locus still requires further research. Thus, the durable resistance locus on 2StS.2J^S^L, which combines multiple resistance genes, represents a strategic genetic resource for wheat breeding programs aiming to enhance disease resistance.

Fluorescence *in situ* hybridization (FISH) and genomic *in situ* hybridization (GISH) serve as powerful tools for chromosomal identification in bread wheat and related species, facilitating the transfer of elite genes from *Thinopyrum* species into the wheat background. Based on the GISH patterns using *Ps. strigosa* genomic DNA (St genome) as a probe, the genomic constitutions of *Th. intermedium* and *Th. ponticum* from different accessions were redesignated [23]; a set of wheat–*Thinopyrum* introgressions could then be successfully detected and applied for wheat breeding purposes. Liu et al. [36] developed the FISH probe pThp12.19 to trace the blue-grain character in a wheat background to *Th. ponticum* chromosome bin 4AgL-6. Yao et al. [37] isolated the *Th. ponticum* genome-specific repetitive sequence B11, which produced specific signals on the entire lengths of *Thinopyrum* chromosomes. Moreover, Li et al. [38] characterized two wheat–*Th. intermedium* double substitution lines X479 and X482 and documented an St-J^S^ translocation by using the J^S^ chromosome-specific repetitive sequence pDb12H. In the present study, we characterized the whole arm translocation between St and J^S^ in Th93-1-6 using Oligo-B11 and Oligo-pDb12H and confirmed the linkage group involved using the Oligo painting system. Moreover, Oligo probe Oligo-6S111 produced strong and distinctly different hybridization signals on terminal regions of the long and short arms, which facilitated the rapid and precise identification of the *Th. intermedium* chromosome 2StS.2J^S^L segments in subsequent irradiated generations. However, the predicted physical locations of probes Oligo-1RS-1 and Oligo-6S111 on chromosomes 2St and 2J^S^ do not match well with the FISH hybridization sites of the 2StS.2J^S^L translocations (Appendix A), indicating that the sequence assembly of the related region in the *Th. intermedium* genome needs to be improved. Therefore, our study has demonstrated that ND-FISH techniques are highly effective for characterizing the translocations among the St, J and J^S^ subgenomes and can be used to precisely distinguish the various alien chromosome segments in the wheat background.

The restricted homoeologous recombination between wheat and its wild relatives creates significant challenges for the development of small-segment translocations containing beneficial alien genes using conventional breeding approaches. To overcome these limitations, radiation mutagenesis and *ph1b* mutant technologies have emerged as powerful tools for generating novel chromosomal rearrangements. These methods not only accelerate the identification of candidate gene regions but also facilitate subsequent gene cloning efforts [39,40]. For example, Han et al. [41] developed a series of deletion and translocation aberrations by crossing irradiated spikes of different triticale lines with a normal wheat cultivar and further cloned two different age-related, powdery mildew-resistant genes, *PmTR1* and *PmTR3*. Li et al. [42] obtained 46 3S^l^#2 recombinants in a population of 1580 BC_1_F_2_ plants derived from a monosomic 3S^l^#2 plant and a homozygous *ph1b* mutant line. Sequential high-resolution mapping and functional verification allowed for the cloning of the powdery mildew resistance gene *Pm13*. In addition, radiation using 60Co-γ rays has been frequently applied in the development of novel translocations between wheat and other Triticeae species such as *Roegneria* [43], *Dasypyrum* [44], *Agropyron* [45] and *Thinopyrum* [46]. In our study, 1040 F_2_ seeds were treated with 60Co-γ irradiation. From 1823 plants of the M_1_ generation, 537 chromosome translocations among the wheat genomes were detected; 37 deletions and translocations involving chromosome 2StS.2J^S^L were obtained. The frequency of translocations between the B and D subgenomes of wheat was the highest, suggesting that repetitive sequence-rich regions may tend to be frequent breakpoints for translocations. Moreover, 12 different types of 2StS.2J^S^L homozygous translocations were recovered from 561 plants of the M_2_-M_3_ generation and used to narrow down the region containing the stripe rust resistance. These results suggest that the mutants of F_2_ seeds from line Th93-1-6 were effectively induced by 60Co-γ rays. Further backcrosses of these heterozygous 2StS.2J^S^L structural aberration lines will be conducted to obtain monosomic individuals to be screened for other potentially useful elite traits.

Stripe rust is a devastating disease that occurs in most wheat-growing regions. Up to now, 86 stripe rust resistance genes (*Yr1*–*Yr86*) have been formally designated [5,6]. Among these, only *Yr50* and *Yr69* have been transferred from *Thinopyrum* species into cultivated wheat [15,47]. In addition, a series of wheat–*Th. intermedium* introgression lines carrying resistance to stripe rust have been given temporary names. Huang et al. [48] developed the wheat–*Th. intermedium* introgression line L693 and mapped a single dominant gene, *YrL693*, against stripe rust on 1BS. Guo et al. [49] created 153 wheat–*Th. intermedium* translocation lines by irradiating with γ rays from 60Co and characterized the stripe rust resistance gene *YrT14* on *Th. intermedium* chromosome 7J or 7J^S^. In the present study, we located the *Yr* gene(s) on chromosome arm 2StS, while wheat plants carrying only the chromosome arm 2J^S^L were susceptible to stripe rust. Moreover, Bansal and Bariana [50] characterized *Yr56* on durum wheat chromosome 2AS bin 2AS5-0.87-1.00. Jiang et al. [51] developed a series of wheat–*Th. ponticum* 2Ae translocation and deletion lines and mapped the stripe rust resistance gene on the FL0.79–1.00 of 2AeS. In the present study, we selected twelve 2StS.2J^S^L deletion and translocation lines from M_1_-M_3_ generations and mapped the stripe rust resistance gene to FL 0.35–0.61 on 2StS, covering the corresponding region of 92.88–155.32 Mb in the reference *Th. intermedium* genome sequence v2.1. The collinearity region of this gene does not overlap with the *Yr* genes mentioned above. Therefore, the novel stripe rust resistance gene on *Th. intermedium* 2StS chromosome from Th93-1-6 was not orthologous to the above-mentioned genes. Specially, three wheat–*Th. intermedium* 2St substitution lines (ES-23, ES-25, ES-26) were characterized using St chromosome-specific molecular markers. A disease resistance evaluation indicated that all three lines exhibited adult-stage resistance to stripe rust [52]. Notably, these lines demonstrated differential responses to *Puccinia striiformis* f. sp. *tritici* races compared to Th93-1-6 containing 2StS, suggesting that these carry distinct stripe rust resistance genes. Furthermore, the 2StS.2J^S^L translocation and deletion lines developed in our study represent valuable genetic resources for transferring potentially elite traits from the 2StS.2J^S^L chromosome segment to common wheat. Our ongoing efforts to create additional homozygous mutants will narrow down the site of the stripe rust locus, thereby accelerating the identification and cloning of candidate resistance genes.

## 4. Materials and Methods

### 4.1. Plant Materials

The wheat–*Th. intermedium* 2StS.2J^S^L (2D) disomic substitution line Th93-1-6, wheat cultivar Mianyang11 (MY11), *Th. intermedium*, Pseudoroegneria spicata and X24C-14 (2J^S^ addition lines) are maintained in our laboratory at the University of Electronic Science and Technology of China. Line Th93-1-6 was derived from the wheat–*Th. intermedium* partial amphidiploid line Th24-19-5. The cross between Th93-1-6 with MY11 was used to conduct the genetic analyses. The seeds of F_2_ generation were irradiated by the Biotechnology and Nuclear Technology Research Institute, Sichuan Academy of Agricultural Sciences, China.

### 4.2. ND-FISH and Oligo-FISH Painting

The root–tip metaphase chromosomes of all germinated seeds were prepared according to the procedure described by Han et al. [53]. The synthetic oligonucleotides Oligo-pSc119.2, Oligo-pTa535, Oligo-B11, Oligo-K288, Oligo-1RS-1, Oligo-6S111, Oligo-pSc200 and Oligo-1123 were used for ND-FISH analysis; their sequences are shown in Appendix A [44,54,55,56,57,58,59]. All oligonucleotide probes were either 5′end-labeled with 6-carboxyfluorescein (6-FAM) for green or 6-carboxytetramethylrhodamine (Tamra) for red signals. After the ND-FISH analysis of metaphase chromosomes, the Oligo-FISH painting technique was performed following the description by Li and Yang [60]. The pictures of FISH results under the Olympus BX-53 microscope were taken with a DP-70 CCD camera.

### 4.3. Molecular Marker Analysis

The DNA of all materials used in this study was extracted from young leaves using the SDS protocol [8]. CINAU markers of linkage group 2, based on intron length polymorphisms, were obtained from Zhang et al. [61]. PCR-based landmark unique gene (PLUG) primers were synthesized according to Ishikawa et al. [62]. EST-PCR primers were obtained from Jiang et al. [52]. Three newly developed 2St-specific primers are listed in Appendix A. The specific 2StS.2J^S^L markers were searched for chromosomal physical locations using the database of *Thinopyrum intermedium* genome V.2.1 (https://phytozome-next.jgi.doe.gov/info/Tintermedium_v2_1, accessed on 21 January 2021). The amplification program involved 4 min at 94 °C; 35 cycles of 45 s at 94 °C; 45 s at 58 °C; 1 min at 72 °C; and a final extension at 72 °C for 10 min. PCR amplification products were analyzed by 1% agarose gel electrophoresis and 8% PAGE gel electrophoresis, as described by Hu et al. [63] and Yu et al. [59], respectively.

### 4.4. Rust Resistance Tests

The reactions of the parent wheat MY11, the wheat–*Th. intermedium* 2StS.2J^S^L (2D) disomic substitution line Th93-1-6 and its derived progenies carrying 2StS telosomes, 2J^S^L telosomes, iso-telosomic 2StS.2StS and 2J^S^L.2J^S^L to stripe rust were assessed at both the seedling and adult plant stages during 2023–2025. Field plots of these lines at the Sichuan Academy of Agricultural Sciences Experimental Station were inoculated with a mixture of stripe rust races CYR32, CYR33, and CYR34. The twelve homozygous 2StS.2J^S^L aberrations were evaluated at the adult plant stage in the 2024–2025 season. Infection types (IT) were scored 14–15 days after inoculation when rust was fully developed on the susceptible check. A 0–4 scale of infection types was recorded according to the standard described by Bariana and McIntosh [30].

### 4.5. Evaluation of Agronomic Traits

During the 2024–2025 growing seasons, the translocation lines C152, the 2StS and 2J^S^L addition lines and their parent line Th93-1-6 and MY11 were grown in Xindu Experimental Station, Chengdu, China. Ten seeds were planted in a row of 1.5 m in length and an inter-row spacing of 0.25 m. Six agronomic traits were evaluated: plant height (PH), tiller number (TN), spike length (SL), spikelets per spike (SS), grains per spike, and thousand-grain weight (TGW). The mean value and standard deviation were calculated by SPSS Statistics19.0 software.

## Figures and Tables

**Figure 1 plants-14-01538-f001:**
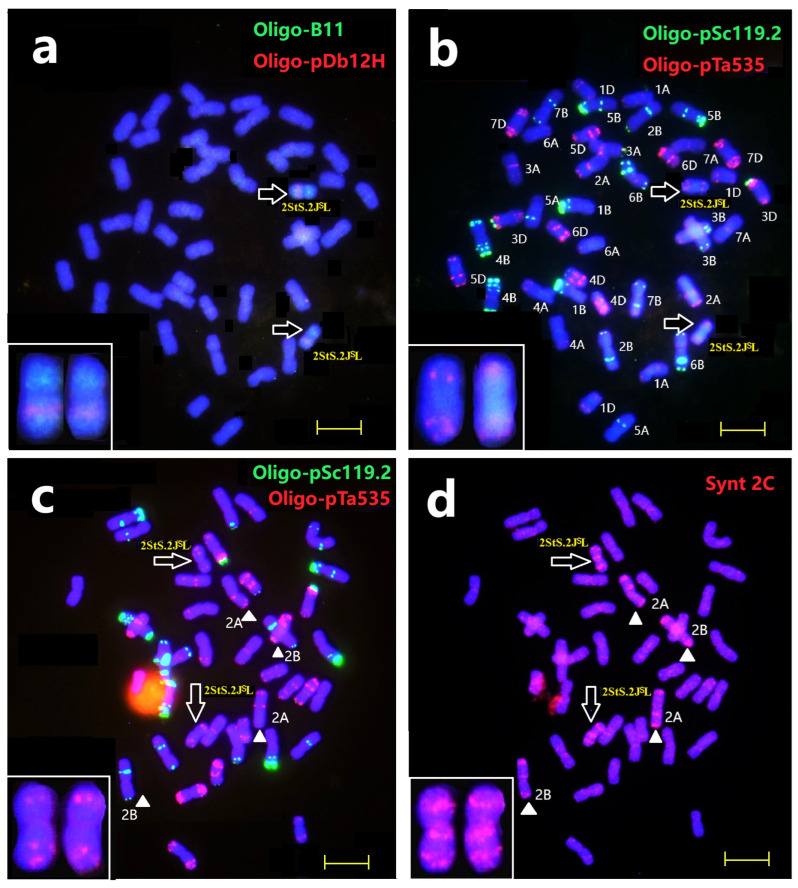
Karyotyping of the mitotic metaphase of Th93-1-6 by sequential ND-FISH and Oligo-FISH painting. Sequential ND-FISH by probes: (**a**) Oligo-pDb12H (red) + Oligo-B11 (green): (**b**,**c**) Oligo-pTa535 (red) + Oligo-pSc119.2 (green); and (**d**) Oligo-FISH by bulk painting with probes Synt2C (red), respectively. White triangles indicate chromosomes 2A and 2B; Arrows indicate chromosomes 2StS.2J^S^L. Bars, 10 μm.

**Figure 2 plants-14-01538-f002:**
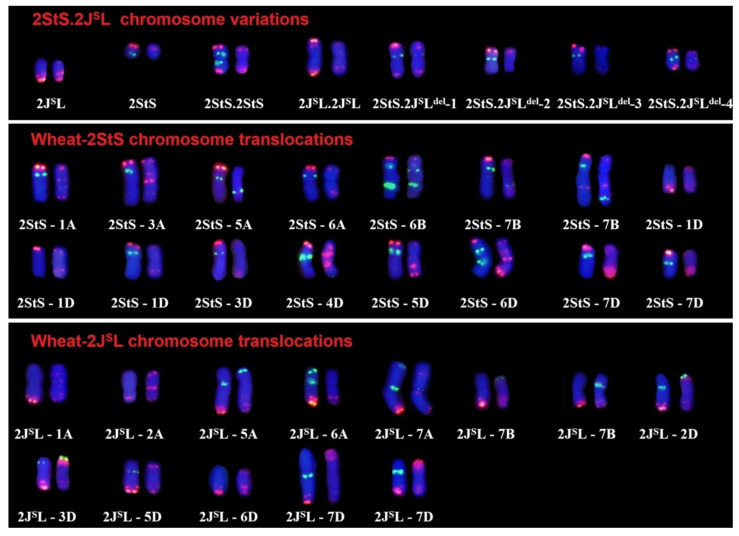
*Th. intermedium* 2StS.2J^S^L chromosome variations from the M_1_ progenies of Th93-1-6 × MY11. The probes were Oligo-6S111 + Oligo-1RS-1 (left) and Oligo-pTa535 + Oligo-pSc119.2 (right).

**Figure 3 plants-14-01538-f003:**
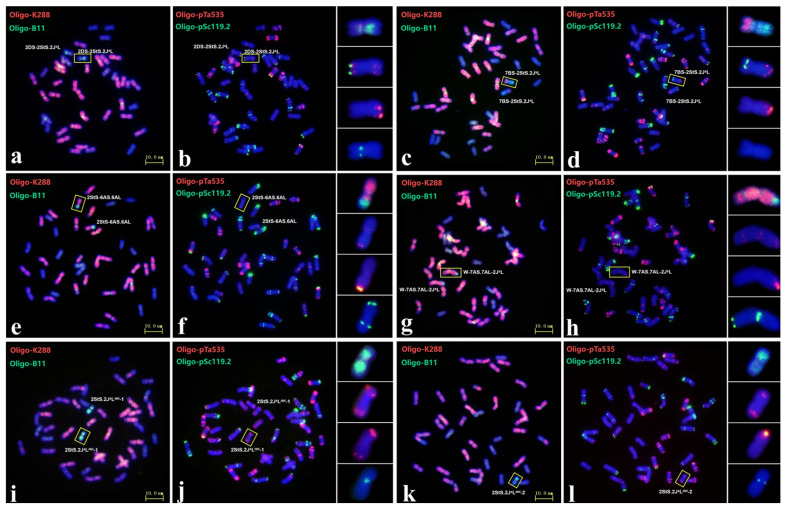
FISH of wheat–*Th. intermedium* 2StS.2J^S^L translocations 2DS-2StS.2J^S^L (**a**,**b**); 7BS-2StS.2J^S^L (**c**,**d**); 2StS-6AS.6AL (**e**,**f**); W-7AS.7AL-2J^S^L (**g**,**h**); deletion lines 2StS.2J^S^L^del^-1 (**i**,**j**); and deletion lines 2StS.2J^S^L^del^-2 (**k**,**l**). The probes were Oligo-k288 + Oligo-B11 (**a**,**c**,**e**,**g**,**i**,**k**) and Oligo-pTa535 + Oligo-pSc119.2 (**b**,**d**,**f**,**h**,**j**,**l**). The corresponding 2StS.2J^S^L aberrations (yellow boxes) are listed on the right; the probes used were Oligo-k288 + Oligo-B11, Oligo-pTa535 + Oligo-pSc119.2, Oligo-6S111 and Oligo-1RS-1, from top to bottom.

**Figure 4 plants-14-01538-f004:**
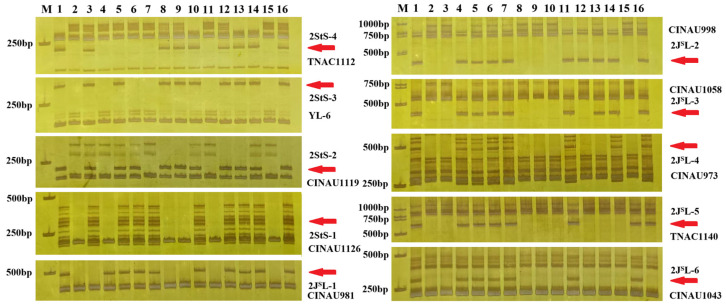
PCR amplification results of representative 2StS.2J^S^L-specifc markers in the twelve translocation and deletion lines: (1) Th93-1-6; (2) MY11; (3) 2StS carrier; (4) 2J^S^L carrier; (5) C651; (6) C671; (7) C923; (8) C42; (9) C152; (10) C797; (11) C796; (12) C423; (13) C788; (14) C269; (15) C115; and (16) C367. Arrows point to chromosome 2StS.2J^S^L-specifc bands.

**Figure 5 plants-14-01538-f005:**
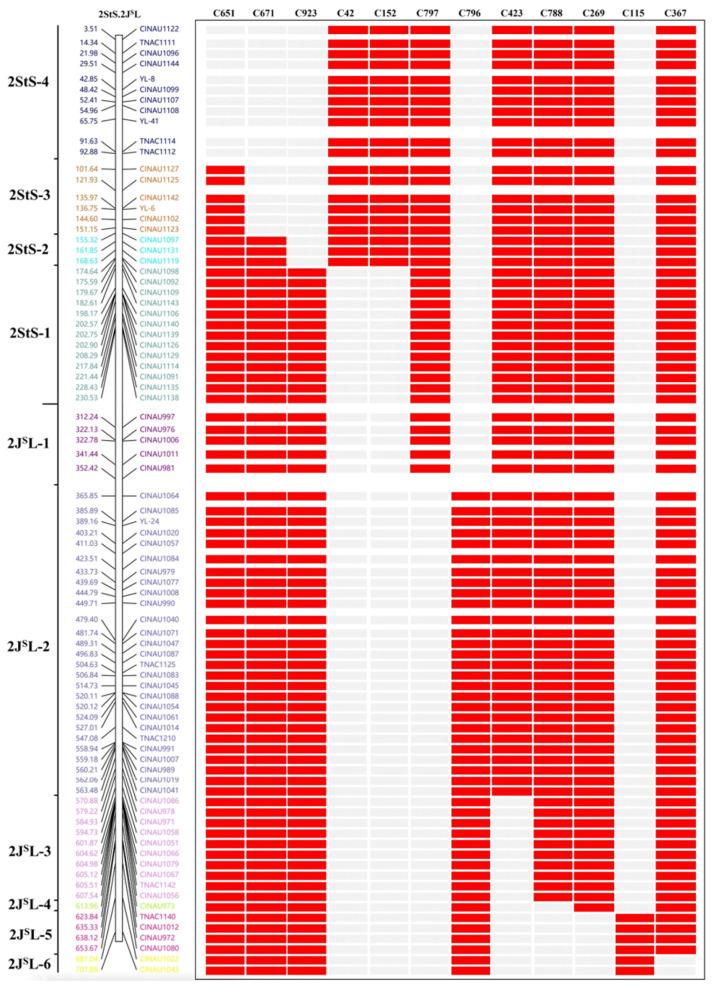
The physical map of chromosome 2StS.2J^S^L. A total of 82 2StS.2J^S^L-specific markers were blasted to determine their locations on the genome sequence of 2StS and 2J^S^L from *Th. intermedium*; different colors indicate the physical regions. The diagram on the right shows different amplification types of 12 2StS.2J^S^L aberrations. The red box represents amplification, while the grey box represents the absence of the 2StS.2J^S^L-specific bands.

**Figure 6 plants-14-01538-f006:**
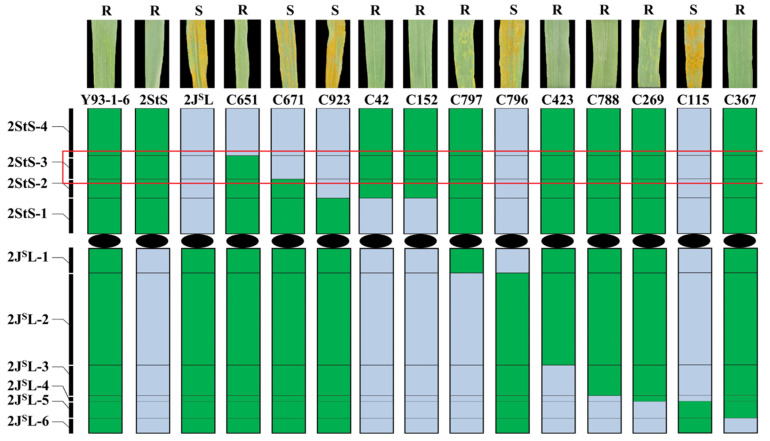
Physical mapping of the stripe rust resistance gene using 12 2StS.2J^S^L aberration lines. Green colors indicate chromosome segments of 2StS.2J^S^L, blue represents wheat chromosome fragments. The red box indicates the physical region of the *Yr* locus.

**Table 1 plants-14-01538-t001:** The agronomic traits of Th93-1-6, MY11, and derived lines.

Materials	Plant Height (cm)	Tiller Number	Spike Length (cm)	Spikelets per Spike	Grains per Spike	1000-Grain Weight (g)
Th93-1-6	89.1 ± 2.0 a	10.6 ± 2.1 a	12.1 ± 0.8 a	24.1 ± 2.1 a	48.9 ± 2.1 a	37.8 ± 0.6 a
MY11	86.4 ± 3.1 b	9.6 ± 1.8 b	10.1 ± 1.0 c	20.6 ± 1.7 c	48.2 ± 1.9 b	35.6 ± 0.5 b
2StS carriers	87.5 ± 2.0 b	10.8 ± 2.2 a	11.3 ± 1.0 b	21.4 ± 2.7 b	48.7 ± 2.1 ab	36.3 ± 0.6 c
2J^S^L carriers	87.6 ± 2.7 b	11.3 ± 1.7 a	9.8 ± 0.8 c	20.4 ± 1.4 c	48.4 ± 2.5 b	35.4 ± 0.5 c
C152	87.3 ± 4.0 b	10.9 ± 2.0 a	11.1 ± 0.8 b	21.4 ± 2.3 b	49.1 ± 2.6 a	37.2 ± 0.7 a

The data indicates means ± standard errors. Different letters indicate significant differences at the *p* < 0.05 level.

## Data Availability

Data are contained within the article.

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
