# Peer review of "Characterization of a New Stripe Rust Resistance Gene on Chromosome 2StS from Thinopyrum intermedium in Wheat"

_plants, 2025, doi:10.3390/plants14101538_

Round 1
Reviewer 1 Report
Comments and Suggestions for Authors
- Line 57. What are the TAI7045 and 78784? Please give the explanation when they were firstly mentioned in the context.
- Lines 101 and 103. What are the Fig. S1a and Fig. S1b? I cannot find the quotation.
- What is the relationship between wheat-Th. intermedium partial amphidiploid line 78784 and Th24-29-5? Is the stripe rust resistance gene on 2St.Js the same as the one on 2Sts in this study? The authors should discuss this in the Discussion section.
- The authors identified 1823 M1 plants by oligo-FISH, the frequencies of translocations between wheat sub-genomes is meaningful data. Please show it in a supplementary table.
- Line 269. The author mentioned a Robertsonian translocation 4StS.1JVS addition line GLA7, please give an explanation about JV genome, is this identical to the Js genome?
- Lines 323-325. Please give an explanation of this conclusion.
- Judging whether or not the stripe rust resistance gene in Th93-1-6 is orthologous to those group-2 derived resistance genes should not only based on their physical position.
There are some mistakes in the manuscript. Please carefully check the English grammar and expression throughout the text.
Lines 34 and 49. 2n = 6x = 42
Line 109. White triangle indicates….; Arrow indicates…
Lines 117-120. This sentence can cause the misunderstanding. 1040 F2 plants or 1040 F2 seeds? and all the F2 seeds or the F3 seeds?
Line 147-148. Since line C671 carried a new translocation instead of 2StS. 2JsL, the sentence “… line C671 had 43 chromosomes, including 41 wheat chromosomes and two 2StS. 2JsL translocation.” will make readers confused.
Line 190. bin 2StS-1
Line 207. Do not start a sentence with Arabic numerals. A total of 82…
Line 356. The abbreviation Pst should be removed in the line 37.
Author Response
Comments 1:Line 57. What are the TAI7045 and 78784? Please give the explanation when they were firstly mentioned in the context.
Response 1:We agree to the correction. We have added the details of the TAI7045 and 78784 in line 58.
Comments 2:Lines 101 and 103. What are the Fig. S1a and Fig. S1b? I cannot find the quotation.
Response 2:Thanks for your suggestion. We modified the quotation in line 101.
Comments 3:What is the relationship between wheat-Th. intermedium partial amphidiploid line 78784 and Th24-29-5? Is the stripe rust resistance gene on 2St.Js the same as the one on 2Sts in this study? The authors should discuss this in the Discussion section.
Response 3:Line 78784 and line Th24-19-5 are wheat-Th. intermedium partial amphidiploid derived from different Th. intermedium accessions. Since the physical regions of the stripe rust resistance gene on 2StS.2JSL (line 78784) have not been determine, we cannot exclude the possibility that the two stripe rust resistance locus are the same. We have added this part in Discussion section.
Comments 4:The authors identified 1823 M1 plants by oligo-FISH, the frequencies of translocations between wheat sub-genomes is meaningful data. Please show it in a supplementary table.
Response 4:Thank you for your reminder. As shown in Fig. S3c, we have showed the frequencies of translocations between wheat sub-genomes.
Comments 5:Line 269. The author mentioned a Robertsonian translocation 4StS.1JVS addition line GLA7, please give an explanation about JV genome, is this identical to the Js genome?
Response 5:Jvs genome is identical to the Js genome in Th. intermedium. Chen et al. (1998) proposed the Th. intermedium genome symbol as JJsS using GISH procedure. Later, Wang et al. (2015) developed EST-SSR primers from recognized ancestral species and inferred the genomes of wild ancestral species of Th. intermedium carrying Dasypyrum repeat sequence. Meanwhile, St2-80 and pDb12h were developed as specific probes of the St genome and V genome, respectively, for use on Th. intermedium (Yang et al. 2006, Wang et al. 2017). Therefore, Js was changed to Jvs, and this genome symbol designation has been accepted by many researchers.
Comments 6:Lines 323-325. Please give an explanation of this conclusion.
Response 6:Lang et al. (2018) predicted that wheat D genome had highest tandem repeat contents, followed by B genome, and A genome was the least. Our previous study on different hybrid combination showed the highest B-D translocation events. Moreover, Ta-3A1 represented the highest copy number of a mini-satellite yet reported of wheat (Lang et al. 2019), and many reported translocations were related to the Ta-3A1 locus (Jiang et al. 2024, Lang et al. 2018, Wang et al. 2023). Therefore, we infer that repetitive sequence-rich regions may tend to be the hot breakpoints for translocations.
Comments 7:Judging whether or not the stripe rust resistance gene in Th93-1-6 is orthologous to those group-2 derived resistance genes should not only based on their physical position.
Response7:Thanks for your suggestion. We made a comparison of the collinearity region of those Yr gene locus to prove our point. We added the sentences in line 351-352.
Comments 8:Lines 34 and 49. 2n = 6x = 42
Response 8:We agree to the correction.
Comments 9:Line 109. White triangle indicates….; Arrow indicates…
Response 9:Thanks for your correction, we have revised them.
Comments 10:Lines 117-120. This sentence can cause the misunderstanding. 1040 F2 plants or 1040 F2 seeds? and all the F2 seeds or the F3 seeds?
Response 10:Thanks for your advice. We have changed the word into “seeds”.
Comments 11:Line 147-148. Since line C671 carried a new translocation instead of 2StS. 2JsL, the sentence “… line C671 had 43 chromosomes, including 41 wheat chromosomes and two 2StS. 2JsL translocation.” will make readers confused.
Response 11:Thanks for your suggestion. We modified the sentence as “line C671 had 41 wheat chromosomes and a pair of terminal translocated chromosomes”.
Comments 12:Line 190. bin 2StS-1
Response 12:Thanks for your correction.
Comments 13:Line 207. Do not start a sentence with Arabic numerals. A total of 82…
Response 13:Thanks for your suggestion.
Comments 14:Line 356. The abbreviation Pst should be removed in the line 37.
Response 14:Thanks for your suggestion. We have deleted the abbreviation.
Reviewer 2 Report
Comments and Suggestions for Authors
Stripe rust is a highly destructive disease that has an important influence on wheat yield. Thinopyrum intermedium is a useful genetic resource for genetic improvement of wheat disease resistance. In this study, a novel wheat-Th. intermedium 2StS.2JSL addition line Th93-1-6 was characterized using ND-FISH and Oligo-FISH. Screening 2,384 M1-M3 progeny (Th93-1-6×susceptible cultivar MY11) by ND-FISH identified 37 structural aberrations of 2StS.2JSL. Cytological bin mapping with 84 molecular markers delineated ten chromosomal bins on 2StS·2JSL, revealing that eight aberration lines containing bin 2StS-3 exhibited enhanced stripe rust resistance. The resistance genes were physically mapped to the 92.88–155.32 Mb region of Th. intermedium chromosome 2StS. These translocation lines provide valuable genetic resources for breeding durable rust-resistant wheat cultivars. However, there were a few major issues in this manuscript.
- Lines 89-95,please make sure the description of the result of FISH is accurate. Figure 1a shows that Th93-1-6 contains 43 chromosomes, while Figure 1b demonstrates three 1D chromosomes.
- In Figure 2, 2StS chromosome displays a green single (left) in 2StS.2JSL chromosome variations. However, two 2StS-1D translocation types lack thissingle in wheat-2StS chromosome translocations. Why do you think they are 2StS-1D chromosome translocations? Additionally, all 2StS.2JSL deletions are lose the fragments of long arm, please the authors explain the reason or possibility.
- Line 129, 37 types of chromosomal aberrations involving 2StS.2JSL were identified. Why did the M2-M3 progenies derived from the 29 2StS.2JSL aberrationsin line140?
- Line 158,it should be “C115” rather than “C155”.
- Lines 375-376,“Oligo-pDb12H” should be described in the part of “Materials and Methods”, and its sequences are also included in Table S1.
- Line 359, there is a punctuation error “..”.
Author Response
Comments 1: Lines 89-95,please make sure the description of the result of FISH is accurate. Figure 1a shows that Th93-1-6 contains 43 chromosomes, while Figure 1b demonstrates three 1D chromosomes.
Response1: Thanks for your correction. We have revised the description of Figure 1a-b.
Comments 2: In Figure 2, 2StS chromosome displays a green single (left) in 2StS.2JSL chromosome variations. However, two 2StS-1D translocation types lack thissingle in wheat-2StS chromosome translocations. Why do you think they are 2StS-1D chromosome translocations? Additionally, all 2StS.2JSL deletions are lose the fragments of long arm, please the authors explain the reason or possibility.
Response 2: Thanks for your question. 1. All the 37 2StS.2JSL variations were detected by probes Oligo-K288 + Oligo-B11 in the first round, and the two 2StS-wheat translocations showed distinct Oligo-B11 signal without Oligo-K288 signal. This results indicated that the two translocations were Composed of alien chromosome and D subgenome chromosome. Sequential ND-FISH with probe Oligo-pTa535 showed that only 1D chromosome contained the relatively weak signal. 2. It is possible that the terminal region of 2JSL contain a large number of repetitive sequence, which may cause the loss of the fragments of long arm.
Comments 3: Line 129, 37 types of chromosomal aberrations involving 2StS.2JSL were identified. Why did the M2-M3 progenies derived from the 29 2StS.2JSL aberrationsin line140?
Response 3: Thank you for your reminder. We forgot the four deletion lines, and we changed number 29 into 33. The remaining four (2StS, 2JSL, 2StS.2StS, 2JSL.2JSL) lines were already homozygous.
Comments 4: Line 158,it should be “C115” rather than “C155”.
Response 4: Thanks for your correction, we have revised it.
Comments 5: Lines 375-376,“Oligo-pDb12H” should be described in the part of “Materials and Methods”, and its sequences are also included in Table S1.
Response 5: Thank you for your reminder. We have added the probe Oligo-pDb12H in the part of “Materials and Methods” and Table S1.
Comments 6: Line 359, there is a punctuation error “..”.
Response 6: Thanks for your correction. We have deleted this punctuation.